

# The running kinematics of free-roaming giraffes, measured using a low cost unmanned aerial vehicle (UAV)

Christopher K. Basu[1], Francois Deacon[2], John R. Hutchinson[1,*] and Alan M. Wilson[1,*]

[1] Structure & Motion Laboratory, Royal Veterinary College, Hatfield, United Kingdom
[2] Faculty of Natural and Agricultural Sciences, Department of Animal, Wildlife and Grassland Sciences, University of the Free State, Bloemfontein, South Africa
[*] These authors contributed equally to this work.

Corresponding author
Christopher K. Basu, cbasu@rvc.ac.uk

## ABSTRACT

The study of animal locomotion can be logistically challenging, especially in the case of large or unhandleable animals in uncontrolled environments. Here we demonstrate the utility of a low cost unmanned aerial vehicle (UAV) in measuring two-dimensional running kinematics from free-roaming giraffes (*Giraffa camelopardalis giraffa*) in the Free State Province, South Africa. We collected 120 Hz video of running giraffes, and calibrated each video frame using metatarsal length as a constant object of scale. We tested a number of methods to measure metatarsal length. The method with the least variation used close range photography and a trigonometric equation to spatially calibrate the still image, and derive metatarsal length. In the absence of this option, a spatially calibrated surface model of the study terrain was used to estimate topographical dimensions in video footage of interest. Data for the terrain models were collected using the same equipment, during the same study period. We subsequently validated the accuracy of the UAV method by comparing similar speed measurements of a human subject running on a treadmill, with treadmill speed. At 8 m focal distance we observed an error of 8% between the two measures of speed. This error was greater at a shorter focal distance, and when the subject was not in the central field of view. We recommend that future users maximise the camera focal distance, and keep the subject in the central field of view. The studied giraffes used a grounded rotary gallop with a speed range of 3.4–6.9 ms$^{-1}$ (never cantering, trotting or pacing), and lower duty factors when compared with other cursorial quadrupeds. As this pattern might result in adverse increases in peak vertical limb forces with speed, it was notable to find that contralateral limbs became more in-phase with speed. Considering the latter pattern and the modest maximal speed of giraffes, we speculate that tissue safety factors are maintained within tolerable bounds this way. Furthermore, the angular kinematics of the neck were frequently isolated from the pitching of the body during running; this may be a result of the large mass of the head and neck. Further field experiments and biomechanical models are needed to robustly test these speculations.

## INTRODUCTION

### Measuring gait parameters outside of the laboratory

Biomechanical measurements of animal locomotion are commonly performed under laboratory conditions. Under these circumstances, confounding variables may be measured and/or controlled. When studying animals, particularly undomesticated animals usually living in natural habitats, the laboratory itself can become a confounding variable. Natural behaviours are less likely to be expressed, and it is difficult to replicate the interactions between the animal and its natural environment (e.g., temperature, light, substrate properties). In many cases it is not logistically possible or safe to study animals in a laboratory setting.

In recent years, the increasing availability of remote sensing has broadened the focus of locomotor research to include more field-based data collection. Accelerometers and Global Positioning System (GPS) devices have been used to derive three-dimensional temporospatial parameters in a variety of human (*Tao et al., 2012*) and non-human locomotor studies (*Daley et al., 2016*; *Hubel et al., 2016*). Whilst these methods are an excellent solution to study locomotor behaviours over an extended period of time, one challenge is that physical access to each study subject is required. This inevitably requires an instance of either manual or chemical restraint.

If two-dimensional (2D) temporospatial kinematics are required, a low-cost alternative is to use an unmanned aerial vehicle (UAV) to gather spatially calibrated video footage of the locomotor behaviour in question. In this study, we demonstrate the utility of a single low-cost UAV in measuring the 2D kinematic gait parameters of free-ranging running giraffes (*Giraffa camelopardalis giraffa*). We use these data to question whether giraffes' running gait is specialised when compared with other mammalian quadrupeds.

### Gait dynamics in giraffes and other quadrupedal mammals

Quadrupeds typically use asymmetrical gaits at faster speeds. In asymmetrical gaits the fore- and hindfeet each act as functional pairs, where each pair of feet can strike the ground simultaneously, or may have a time lag between the footfall of the left and right side, in which case there is a leading and a trailing foot. Galloping gaits may be further defined by the pattern of the leading limbs. In a transverse gallop, the leading limbs of the fore and hind quarters are on the same side of the body, *versus* a rotary gallop where the leading limbs are on the opposite sides of the body. In either case, both a gathered and extended aerial phase can be present, where all feet are airborne (*Biancardi & Minetti, 2012*; *Hildebrand, 1977*).

At walking speeds, giraffes use a lateral sequence walk, which is dynamically similar to the slow gaits of other mammalian quadrupeds (*Basu, Wilson & Hutchinson, 2019*). The theory of dynamic similarity predicts that geometrically similar animals move with similar dimensionless stride parameters at equivalent dimensionless speeds. Dimensionless speed is expressed here as Froude number (Eq. (1)), where $u$ = speed (ms$^{-1}$), $h$ = shoulder height (m) and $g$ = acceleration due to gravity (9.81 ms$^{-2}$).

$$Fr = \frac{u^2}{gh}.\tag{1}$$

Giraffes appear to diverge from the predictions of dynamic similarity at faster than walking speeds. Based on observations of other mammalian quadrupedal gaits (*Hildebrand, 1976*) and predictive modelling of quadrupedal footfall sequences (*Cartmill, Lemelin & Schmitt, 2002*), giraffes are expected to select a running pace as their intermediate gait. A study involving simulations of quadrupedal gaits also suggested that giraffes will select a pacing gait at intermediate speeds (*Suzuki et al., 2016*); however this model inaccurately predicted that giraffes use a diagonal sequence walk at slow speeds, contrasting with the experimentally observed lateral sequence walk (*Basu, Wilson & Hutchinson, 2019*). Giraffes instead seemingly transition consistently from a walk to a rotary gallop (*Dagg & Vos, 1968*; *Maxwell, 1924*). The restricted choice of gait is in contrast to most other cursorial quadrupeds (*Heglund & Taylor, 1988*; *Hildebrand, 1976*), but not exclusive to giraffes; for example elephants use the lateral sequence walk across their entire speed range (*Hutchinson et al., 2006*). At near-maximal running speeds, giraffes are thought to use lower mean stride frequencies (and consequently higher stride lengths) than is expected for an ungulate of their body mass; an ability that may be facilitated by their elongate limbs (*Alexander, Langman & Jayes, 1977*).

Giraffes' long neck may have functional consequences with respect to the locomotor system. Evolutionary elongation of giraffes' cervical vertebrae has effectively lengthened their horizontal axis (*Badlangana, Adams & Manger, 2009*). In other galloping quadrupeds, the horizontal axis of the skeleton is dynamic, where fluctuations in neck angle and body pitch occur during walking and running (*Dunbar, 2004*; *Dunbar et al., 2008*). Such angular fluctuations may serve to stabilise these axial body segments in world space. In giraffes, *Dagg (1962)* noted the periodic angular fluctuations of the neck, and found this to be larger in magnitude during the gallop than in the walk. One way to define this effect of neck and pitching angles on the horizontal axis is to determine the phase relationship between the kinematics of the trunk and the neck (*Basu, Wilson & Hutchinson, 2019*).

Our aims in this study are to (1) validate the use of a UAV in measuring temporospatial gait parameters and suggest recommendations for optimising data quality; (2) determine which running gait(s) the giraffes select across their speed range, (3) determine how stride parameters change within the running gait, (4) assess whether giraffes' running gait is specialised compared to other cursorial quadrupeds, and finally (5) measure the angular kinematics of the neck and body, and predict how their phase relationship contributes to body segment stability.

## MATERIALS & METHODS

Video data of giraffes' running gait were recorded from three field sites in the Free State Province, South Africa. A total of 35 giraffes were available for study; these varied in age, size and sex (Table 1). The experimental protocol varied between the field sites, and was dependent on the giraffes' degree of familiarity with people. A Phantom 4 UAV (DJI, Guangdong, China) was used to film giraffes' locomotion from a lateral viewpoint, at 120 Hz, $1,920 \times 1,080$ pixel resolution, with a 20 mm lens. This was the maximum possible frame rate and image resolution of any low-cost UAV (<£1,000) at the time the study was conducted.

Table 1 Details of study sites.

|  | Size (hectares) | Number of giraffes | Giraffe temperament | Calibration methods used |
|---|---|---|---|---|
| Site 1 | 460 | 2 | Tame | A,B,C |
| Site 2 | 250 | 6 | Wild | C |
| Site 3 | 12,500 | 27 | Wild | C |

The giraffes were motivated to run using different methods. In field site 1 (Table 1), giraffes were accustomed to following a vehicle as part of their usual routine. During data collection, a vehicle was driven along a straight track at steady speed. A 200 m segment of this track was outlined with white paint marks spaced at 2 m intervals. This speed of the vehicle was periodically varied to induce different steady state running speeds. In field sites 2 and 3, the giraffes were less habituated to humans and vehicles; in these sites the sound and proximity of the UAV was sufficient to induce galloping for short distances.

## Ethical statement

This study had ethical approval from both the Royal Veterinary College (URN 2016 1538) and the University of the Free State, South Africa (UFS-AED2016/0063). A regional permit to study giraffes was obtained from the Department of Economic Development, Tourism and Environmental Affairs (DEDTEA, Free State Province, South Africa; permit number 01/34481). Data were gathered during a two week period in October 2016; that month was chosen as it was during the dry season, and typically lacks the extreme low and high temperatures seen at other times of the year. Measures were taken to minimise stress and danger to the giraffes, and management personnel were present at each site at all times. Firstly, data were gathered during cooler times of day to minimise the risk of heat stress. Secondly, giraffes were only filmed in open habitats with minimal ground obstructions. Thirdly, individual giraffes were only motivated to run up to twice daily, and for sustained periods of time less than one minute. In between bouts of data collection, giraffes were allowed to express normal behaviour.

## Video calibration

Prior calibration of UAV mounted cameras is not possible, as the subject-to-camera distance is not constant as it is with a static camera. Although this UAV model had proximity sensors, these were not suitable to measure the subject-to-camera distance. Three calibration methods were used, either in combination or isolation (Table 1). With each method, the distance from the metatarsophalangeal joint to the most caudal point of the calcaneus (MTP-C; Fig. 1) was used to calibrate each frame of digitised video. The MTP joint centre was measured as the centroid of a circle drawn around the joint. Video footage was manually digitised using the DLTDV6 (*Hedrick, 2008*) script for Matlab (MathWorks Inc., Natick, MA, USA) software, using a system of virtual markers (*Basu, Wilson & Hutchinson, 2019*). The giraffes' natural coat patterns were used to maximise digitisation repeatability. Digitised points were filtered using a zero-phase 4th order Butterworth filter with a 6 Hz cut-off.

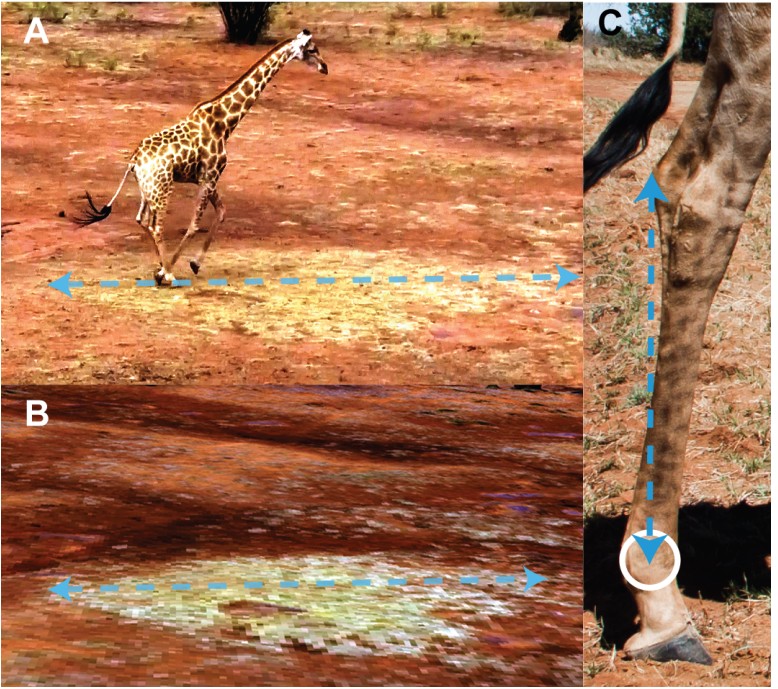

**Figure 1** **A comparison of features from UAV footage, a 3D textured mesh and a photograph.** (A) Still frame of a galloping giraffe from UAV video. (B) Render from a textured 3D mesh, created from aerial photographs. The natural feature from the still image can be referenced to the textured 3D mesh. The feature was measured in the same plane as the hindlimb, and this measurement used to convert the MTP-C pixel distance (C) into an estimation of the metric distance. Image source credit C Basu.

The MTP-C distance was measured using method A, B or C. Method A used close range digital photography and a trigonometry calculation (Eqs. (2)–(4)) to estimate the size of MTP-C. Photographs (Canon EOS 500D, 18–200 mm 5.6f lens, Tokyo, Japan) were opportunistically taken of each giraffe, focusing on the hindlimb. For each photograph, the distance from the camera sensor to the subject's area of interest was measured using a laser rangefinder (Disto D5, Leica, Wetzlar, Germany), and the lens focal length was later retrieved from the image metadata.

$$\text{horizontal angle of view } (°) = 2 * \text{atan} \left( \frac{\text{sensor size}}{2 * \text{focal length}} \right) \tag{2}$$

$$\text{horizontal field of view (m)} = 2 * \text{distance} * \tan \left( \frac{\text{angle of view}}{2} \right) \tag{3}$$

$$\text{photo calibration constant} = \frac{\text{horizontal field of view (m)}}{\text{number of horizontal pixels}}. \tag{4}$$

The pixel MTP-C distance was then measured in each photograph using ImageJ (*Rasband, 2009*), and converted to estimated metric distance using the calibration constant from Eq. (4). This trigonometry method assumes that the camera sensor is perpendicular to the subject. It was not possible to quantitatively measure the optical axis of the camera;

therefore, parallax errors may have affected the resulting calculations. To offset this error, the mean estimate was used to calibrate video footage. Due to the close-range nature of the photographs, this method was only suitable in field site 1, which had a more controlled and predictable environment.

In Method B, 2 m ground markers were used to calibrate 10 still frames from video footage of each giraffe, as they moved parallel with the markers. MTP-C distance was estimated from these frames, and the mean value used to calibrate subsequent video data. This method was only possible in field site 1, where the giraffes could be led along the 200 m track.

Method C was used where the giraffes' locations and locomotion trajectories could not be anticipated (field sites 2 and 3). After recording video data, a textured 3D mesh of the terrain was produced using aerial photographs taken from the UAV during an aerial survey at 40 m altitude. The derived models were created using the software package Pix4D (Lausanne, Switzerland), which takes GPS stamped photographs as an input, and outputs scaled and textured 3D meshes of the corresponding terrain, using a photogrammetry method. These models were used to identify and measure prominent ground features, which could be cross-referenced with features in digitised video, and subsequently used to measure MTP-C distance. The criteria for suitable ground features were that they had to be within the sagittal plane of the giraffe, and be distinguishable on the 3D mesh and the video (Fig. 1).

## Kinematic data processing

Footage from statically mounted videos has a fixed coordinate system, originating from one corner of the video. Footage from a moving camera has no fixed coordinate system, as the boundaries of the video change with each frame. To compensate for this, an in-plane static ground point was digitised for each analysed stride, and all coordinates were translated relative to this fixed point. An additional static ground point was digitised to measure the tracking error associated with digitisation. The error was taken as the standard deviation of the point's coordinates around its mean location. Rotational transformation of digitised points was not performed, as camera rotation during flight was corrected by a three-axis stabilised gimbal, which corrects the pitch, yaw and roll of the camera to the nearest 0.02° (https://www.dji.com/uk/phantom-4/info).

This experimental setup allowed for a single lateral camera view. A consequence was that far side foot-on events frequently were obscured; however far side foot-off events reliably were visible. These data allowed contralateral limb phase (the lag between a pair of fore or hindlimb footfalls) to be measured, however ipsilateral limb phase (the lag between fore and hind footfalls) could not be measured, as this requires the foot on and off timings for all four feet to be measured (Hildebrand, 1977). The complete set of foot events were visible in one stride, which was used to quantify the footfall sequence for the galloping gait. We defined a stride by the timing from the nearside hindlimb foot-ground contact event, to the timing of the subsequent nearside hindlimb foot event.

A number of criteria were used to ensure that stride data were suitable for analysis. An assumption of linear regression is that the data units are independent from each other. To

ensure that this assumption was met, only one stride from a sequence of consecutive strides was used in the analyses. A sequence of consecutive strides was defined as being bounded by either a change in gait, or an obvious change in steady state speed. Only steady-state strides were analysed; strides which featured a 20% or greater change in speed over their course were excluded. Speed was subsequently measured as the horizontal displacement of the giraffes' withers over the course of one stride (m), divided by the time interval (s).

Strides were manually segmented. Ideally a velocity threshold method provides a repeatable method of detecting foot contact events (*Starke & Clayton, 2015*), but could not be used in this instance due to excessive measurement noise. A custom-written Matlab script then applied the pixel calibrations and transformations to the raw data. The following parameters were calculated: running speed ($ms^{-1}$), stride length (m), stride frequency (Hz), footfall timings, contralateral limb phase (the fraction of the stride between footfalls of leading and non-leading limbs), stride duration (s), stance duration (s), duty factor (the fraction of the stride that a given foot is in contact with the ground), neck angle and body pitch angle (°).

The phase relationship between neck angle and body pitch was calculated as the percent congruity. This is a measure of how often the slopes of neck angle and body pitch time series share the same sign; indicating whether the respective angular patterns of the neck and body are in-phase (high congruity) or out-of-phase (low congruity) (*Ahn, Furrow & Biewener, 2004*). Using the angle convention defined in Fig. 2A, 100% congruity represents simultaneous neck ventroflexion and upward 'motor-bike' body pitching, whilst 0% represents simultaneous neck dorsiflexion with upward body pitching.

## Comparisons with dynamic similarity

Power equations were determined to fit Froude number to relative stride length (stride length/leg length), fore duty factor and hind duty factor. Plotted data from Figs. 3 and 4 of *Alexander & Jayes (1983)* were digitised using a web-based application (https://automeris.io/WebPlotDigitizer/), and power curves were fitted to the resulting data, to test how well our giraffe data fit those data for other mammals. The 95% confidence intervals of the exponents and coefficients from the current dataset were compared with the corresponding intervals from Alexander and Jayes' models of dynamic similarity (*Alexander & Jayes, 1983*).

## Statistical analysis of stride parameters

Statistical procedures were carried out using the Matlab Statistical Toolbox. All stride parameters were tested for normality using a Kolmogorov–Smirnov test. Differences in fore or hindlimb parameters were identified using a two-tailed $t$-test, and analysed separately if statistically significant differences were present. OLS linear regressions were performed using speed as the independent variable, and stride and force parameters as the dependant variable. To compensate for multiple statistical comparisons, critical p-values were adjusted using the Benjamini–Hochberg procedure, using a false discovery rate of 0.05 (*Benjamini & Hochberg, 1995*).

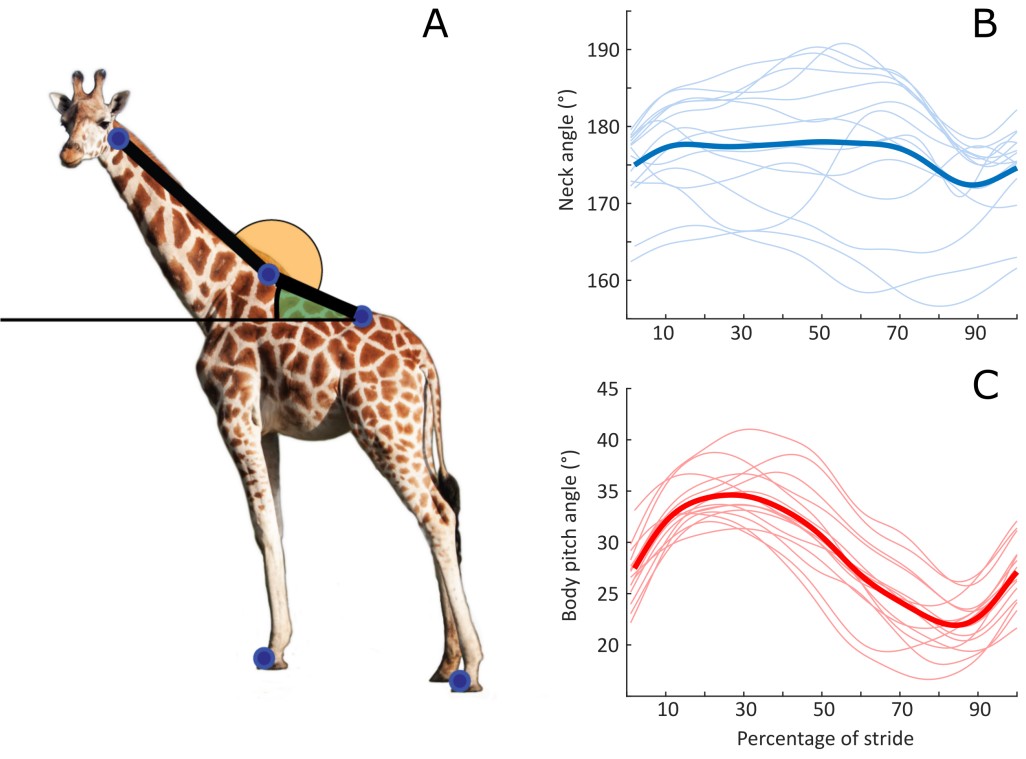

**Figure 2  Neck and pitch angle time series during steady state rotary galloping.** (A) The anatomical definitions of neck angle and body pitch, demonstrated in a standing individual. Image source credit C Basu. Neck angle (B) and body pitch (C) time series from strides commencing with foot-ground contact by the non-leading hindlimb, with mean time series (thicker line).

## Method comparison and validation

The precision of speed measurements was dependent on the precision of the MTP-C measurements, i.e., the calibration method. The three calibration methods were compared in field site 1, using one giraffe. In the case of method C, MTP-C distance was estimated separately using both the artificial ground markers ($C_{ARTIFICIAL}$), and naturally occurring features ($C_{NATURAL}$).

Method A (using close range photography) subsequently had the lowest standard deviation between ten repeated measurements, and was used to quantify the percentage error of the other methods:

$$\text{Percentage error of Method}_{\text{OTHER}} = \frac{\text{Estimate}_{\text{OTHER}} - \text{Estimate}_{\text{A}}}{\text{Estimate}_{\text{A}}} * 100. \tag{5}$$

We assessed the accuracy of UAV derived measurements of speed in a separate validation study, based on Method A. The field conditions were approximated by measuring the speed of a human subject as they ran on a treadmill. Written consent from the human participant was obtained. Prior to the experiment, the subject's knee to ankle distance was measured using close range photography and Eqs. (2)–(4). Skin markers overlying the lateral femoral condyle and lateral malleolus were used to aid digitisation, and the treadmill belt was marked in 0.5 m increments. The speed of the treadmill belt was used as the 'gold standard'

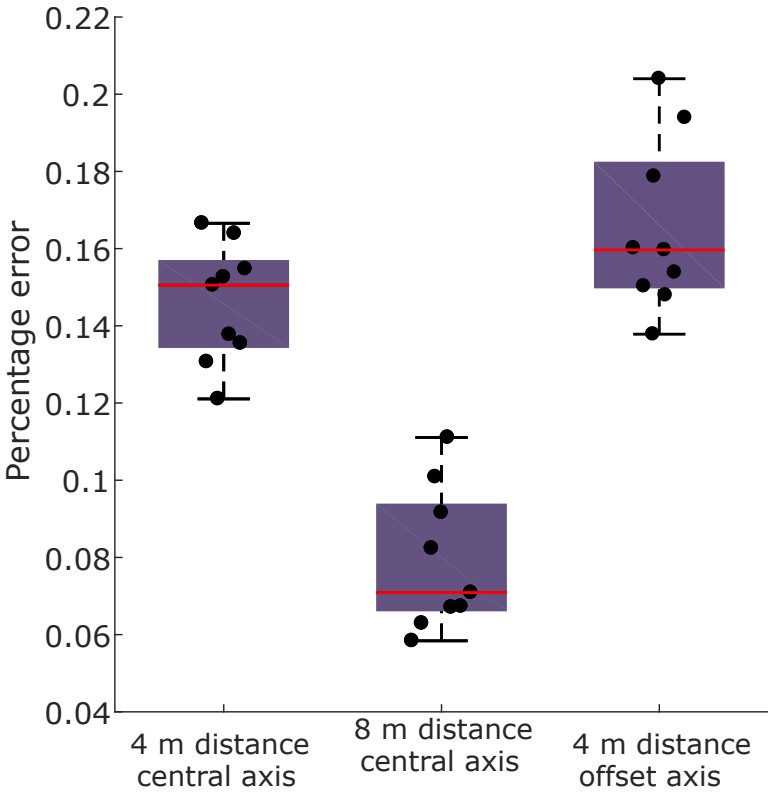

**Figure 3** Box and whisker plot showing percentage error of UAV derived speed measurements compared to treadmill speed, with overlying data points. Human running speed measured by the UAV was most accurate when the subject was furthest distance, and when they were centred in the field of view.

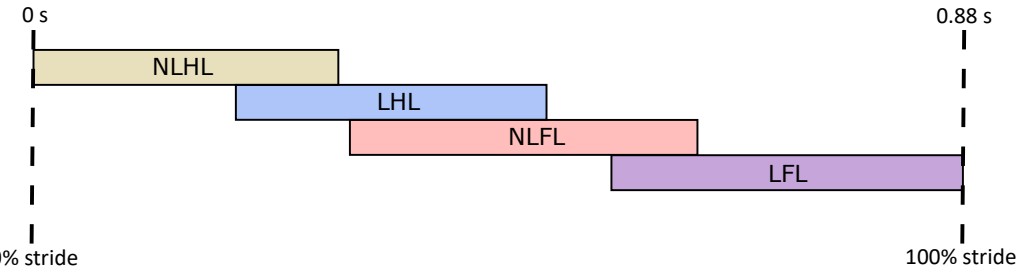

**Figure 4** Footfall sequence of a representative stride from a giraffe running at 4.88 ms⁻¹, with a forelimb duty factor of 0.39 and hindlimb duty factor of 0.38. NLHL, non-leading hindlimb; LHL, leading hindlimb; NLFL, non-leading forelimb; LFL, leading forelimb.

to which UAV derived speeds were compared. Treadmill speed was measured using marker displacement (m) divided by time (s). Subject speed was separately measured as in Method A, using a point on the subject's chest and the treadmill belt markers to measure displacement. Each frame of video data was calibrated using the subject's knee to ankle distance.

We performed trials under three conditions, where the focal distance and camera axis were independently varied. In each condition, the subject was instructed to run through their comfortable speed range on a treadmill. The treadmill speed was adjusted accordingly in stages. In the first condition the Phantom 4 UAV was manually held (i.e., without a fixed support) at a distance of 4 m from the subject, and for the second condition at 8 m distance. In both these conditions the camera's axis was centred on the subject. In the third condition, the focal distance was kept at 4 m, but the camera's focal axis was offset laterally so that the subject was recorded in the lateral third of the camera's field of view. This final condition explicitly tested the effect of optical error (e.g., parallax error, radial distortion) on data accuracy. Percentage error ($\text{Speed}_{\text{ERROR}}$) was defined from the gold standard measurements of speed ($\text{Speed}_{\text{GOLD}}$) and measurements derived from the UAV ($\text{Speed}_{\text{UAV}}$):

$$\text{Speed}_{\text{ERROR}} = \frac{\text{Speed}_{\text{GOLD}} - \text{Speed}_{\text{UAV}}}{\text{Speed}_{\text{GOLD}}} \times 100. \tag{6}$$

We then tested the effects of camera focal distance and axis offset on percentage error, using a two-way ANOVA.

## RESULTS

Over 50 stride sequences were filmed, where the body and footfalls of at least one giraffe where clearly visible. Data from 25 representative strides from four similarly-sized adult individuals were included in the analysis (e.g., Video S1). The mean digitisation error across all trials was ±1.6 pixels, with a resulting mean calibrated error of ±0.01 m.

### Method comparison and validation

Close range photography (Method A) yielded MTP-C estimates with the lowest standard deviation (Table 2), and was used to estimate percentage error for Method B and C. The percentage error of method B was 3.7%, with a 4 mm higher standard deviation than Method A. When Method C (using a GPS calibrated terrain model) was applied using the artificial markers, the percentage error was also 3.7%. The percentage error (5%) and standard deviation (0.1 m) was larger when natural ground features ($C_{\text{NATURAL}}$) were used. Method A was subsequently used to calibrate footage from field site 1, and method $C_{\text{NATURAL}}$ was used for sites 2 and 3.

We compared speeds measured using a UAV with a gold standard method, using a human running on a treadmill. Speeds measured with the UAV were consistently lower than treadmill speed (Fig. 3). Across all the experimental conditions, the mean measurement error was 13% (±5% standard deviation; SD) of treadmill speed; i.e., UAV measured speed was 13% lower than treadmill speed. The condition with the highest mean error was condition 3 (focal distance of 4 m with an offset axis) with an error of 17% (±2% SD); and the lowest was condition 2 (focal distance of 8 m), with an error of 8% (±2% SD). Both camera focal distance and axis offset resulted in significant differences in $\text{Speed}_{\text{ERROR}}$ (ANOVA $p < 0.0001$ and $p = 0.04$ respectively), with distance having the largest effect (Fig. 3).

**Table 2  Comparison of MTP-C distance estimates from one giraffe in field site 1.** Method A resulted in the lowest standard deviation, and was used as the standard to which the other methods were compared. Method C was compared twice; once using artificial ground markers, once using naturally occurring features.

| Method | MTP-C distance estimate | | | |
| --- | --- | --- | --- | --- |
| | Range (m) | Mean (m) | Standard deviation (m) | % error |
| A | 0.78–0.84 | 0.814 | 0.018 | |
| B | 0.76–0.82 | 0.783 | 0.022 | 3.7 |
| $C_{natural}$ | 0.74–0.88 | 0.850 | 0.099 | 4.9 |
| $C_{artificial}$ | 0.76–0.83 | 0.784 | 0.024 | 3.7 |

## Giraffe running kinematics

The observed speeds ranged from 3.4 to 6.9 ms$^{-1}$, with a mean of 5 ms$^{-1}$ (or Fr 1.35). Given that the individuals studied were of similar size, absolute speeds were analysed. Giraffes moving immediately slower than this speed range used the lateral sequence walking gait, consistent with previous observations in giraffes (*Basu, Wilson & Hutchinson, 2019*). In the adult giraffes studied, the observed running gait was a grounded rotary gallop (Fig. 4). Brief aerial phases were only observed in juveniles, and are not covered in the present analysis.

Linear regression results are reported in Table S1. A statistically significant increase in stride length ($p < 0.001$) and stride frequency ($p < 0.001$) was observed (Fig. 5), respectively representing an increase of 0.5 m and 0.05 Hz for each 1 ms$^{-1}$ increase in speed. Stance duration (Fig. 6A) was greater in the forelimb compared with the hindlimb ($p < 0.001$), and decreased with speed ($p < 0.001$). In contrast, swing duration (Fig. 6B) was shorter in the forelimb (blue) versus the hindlimb (red) ($p < 0.001$). There was however, no observed relationship between swing duration and speed ($p = 0.8$). Duty factors (Fig. 6C) and contralateral limb phase (Fig. 7) decreased with speed ($p < 0.001$ and $p = 0.002$ respectively), and were greater in the forelimb ($p < 0.001$ for both parameters).

The body pitch fluctuation resembled a sine wave, and cycled once throughout the stride (Fig. 2C), with an increase in pitching coinciding with the foot-off events of the forelimbs. The neck angle oscillated once throughout the stride, although the pattern of change was more irregular and more variable between strides (Fig. 2B) than the body pitch angle. Neck range of motion and body pitching did not vary as a function of speed ($p = 0.68$ and 0.07 respectively). Neck angle and body pitch had a mean percent congruity of 70% (standard deviation 18%).

## Dynamic similarity

Table 3 summarises the equations that describe how relative stride length and duty factors changed with dimensionless speed in running giraffes. The coefficients and exponents were compared with the predictive equations for dynamic similarity (*Alexander & Jayes, 1983*). Relative stride length in giraffes was consistent with these predictions. The coefficients (a) describing 'duty factor versus speed' in giraffes were significantly lower than expected from the models for dynamic similarity.

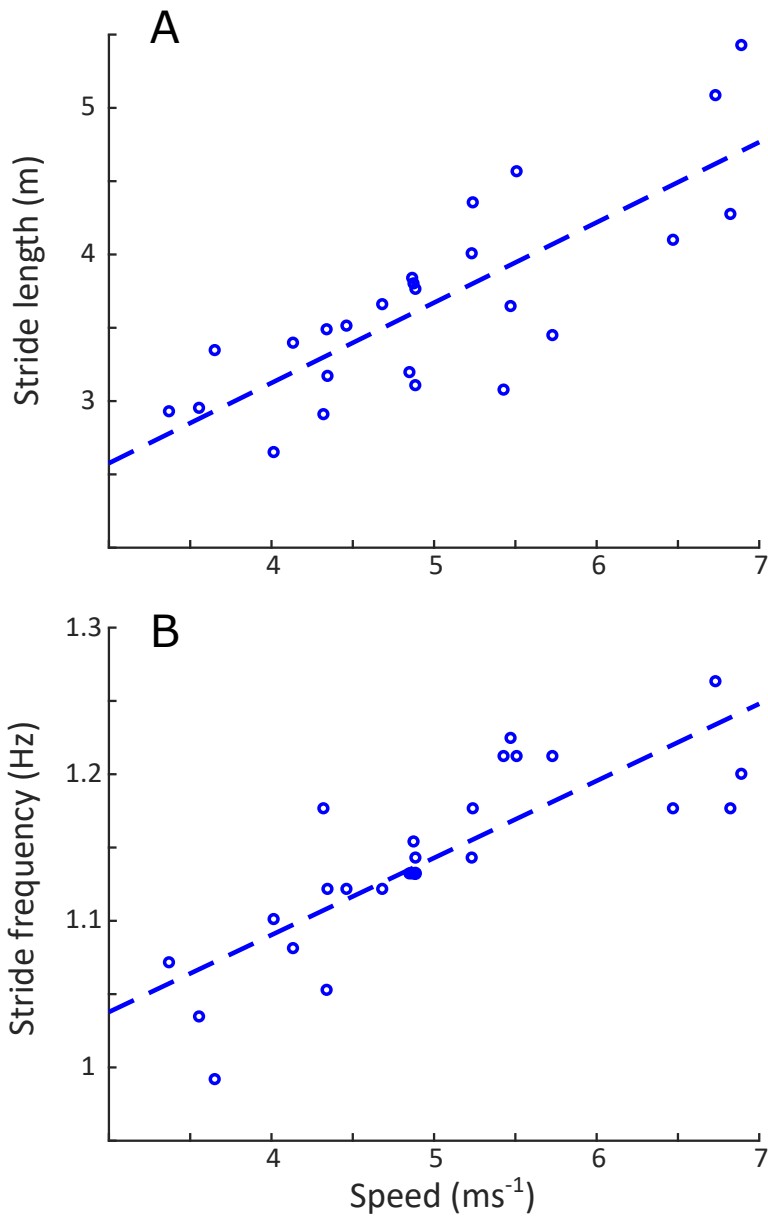

**Figure 5  Change in stride length (m) and stride frequency (Hz) as a function of speed (ms$^{-1}$).** Stride length (A) changed by 0.55 m per unit of speed ($u$): $y = 0.55u + 1.43$, $R^2 = 0.63$, $p < 0.001$. Stride frequency (B) changed by 0.05 Hz for every unit of speed ($u$): $y = 0.05u + 0.88$, $R^2 = 0.66$, $p < 0.001$.

## DISCUSSION

This study has highlighted the potential gains of using a UAV to collect field-based kinematics. Using a moving *vs.* static camera allowed for a larger quantity of data to be recorded than would otherwise have been possible. We have shown that speed and other 2D kinematic parameters can be measured in this way. The major technical challenge was calibrating the raw footage to calculate spatial parameters. The most consistent calibrations

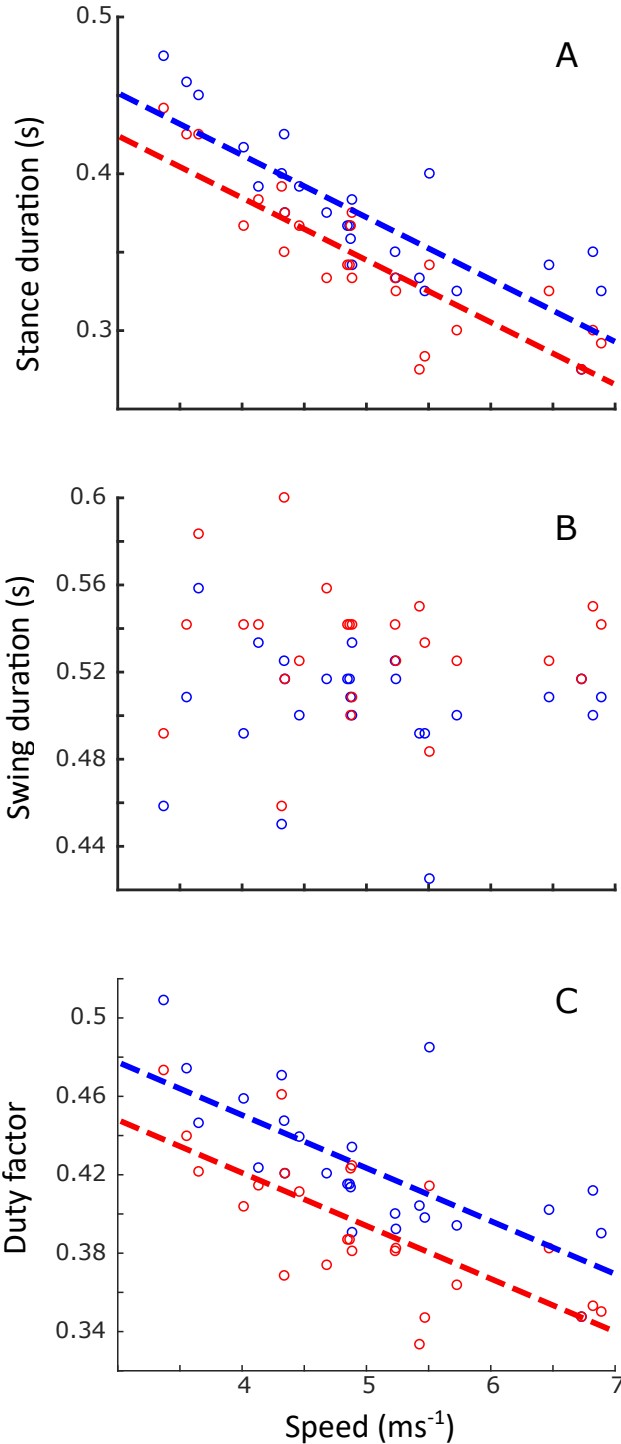

**Figure 6 Changes in stance duration (s), swing duration (s) and duty factor as a function of speed (ms⁻¹).** (A) Stance duration was longer in the forelimb (blue circles) than the hindlimb (red circles), and both decreased with speed (FL: $y = -0.04u + 0.57$, $R^2 = 0.70$, $p < 0.001$; HL: $y = -0.04u + 0.55$, $R^2 = 0.74$, $p < 0.001$). Swing duration (B) was independent of speed, resulting in a duty factor (C) which was greater in the forelimb than the hindlimb, and which decreased with speed (FL: $y = -0.03u + 0.55$, $R^2 = 0.49$, $p < 0.001$; HL: $y = -0.03u + 0.53$, $R^2 = 0.54$, $P < 0.001$).

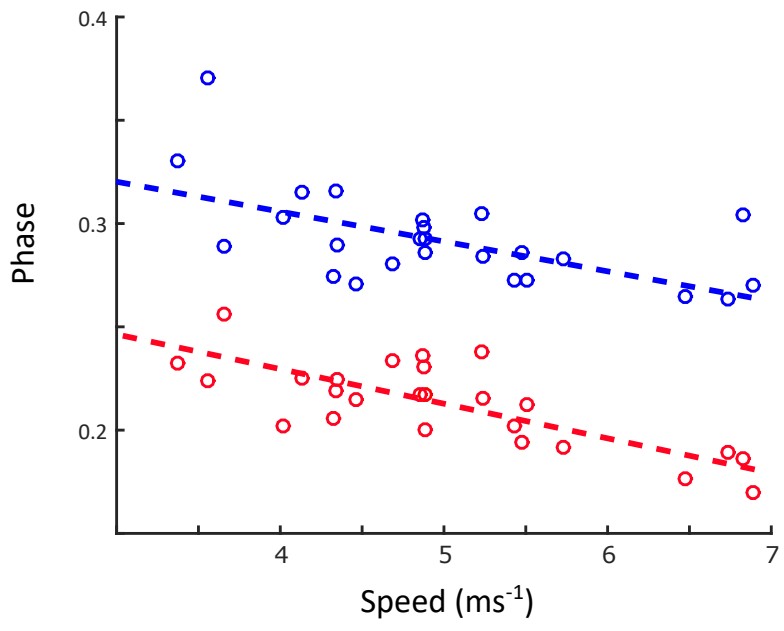

**Figure 7 Changes in contralateral limb phase as a function of speed (ms$^{-1}$).** Contralateral limb phase is expressed as the relative lag time between footfalls of the left and right side. Phase was greater in the forelimb than the hindlimb, and decreased with speed. A decrease in phase indicated that the overlap between left and right footfalls was greater at faster speeds (FL: $-0.01u + 0.36$, $R^2 = 0.35$, $p = 0.002$; HL: $y = -0.02u + 0.29$, $R^2 = 0.57$, $p < 0.001$).

**Table 3 Comparison of power equations from the current dataset, with equations for dynamic similarity.** Data from *Alexander & Jayes (1983)* were digitised, with the exception of hind duty factor (*) which was not presented as a figure. Equations are in the form $y = a(Fr)^b$. The ±95% confidence interval is given in brackets where available. Relative stride length (stride length/leg length) in giraffes was consistent with the predictions for dynamic similarity; i.e., giraffes take proportionally similar strides. Fore and hind duty factors in giraffes were lower than predicted by dynamic similarity, as indicated by the significantly lower coefficient (a).

| Stride parameter | Giraffe coefficients | | Coefficients from *Alexander & Jayes (1983)* | |
|---|---|---|---|---|
| | a | b | a | b |
| Relative stride length | 1.98 (0.11) | 0.31 (0.11) | 1.85 (0.09) | 0.43 (0.02) |
| Fore duty factor | 0.44 (0.01) | −0.17 (0.06) | 0.52 (0.02) | −0.27 (0.04) |
| Hind duty factor | 0.41 (0.01) | −0.19 (0.07) | 0.53[*] | −0.28 (0.03)[*] |

were gained when the study subject was close enough to take repeated photographs, or when artificial markers were included in the field of view. We tested the accuracy of this UAV method, and found that speed accuracy was optimised at a longer focal distance, and when the subject was centred in the field of view (Fig. 3). Both of these conditions minimise the effect of optical error on spatial measurements (distances and angles). Optical error is expected to minimise to a plateau with subsequent increases in subject focal distance, or as the subject converges on the camera's optical axis (*Bräuer-Burchardt, 2007*; *Duane, 1971*; *Kirtley, 2006*). Long focal distances are also preferred to minimise potential stress to animal

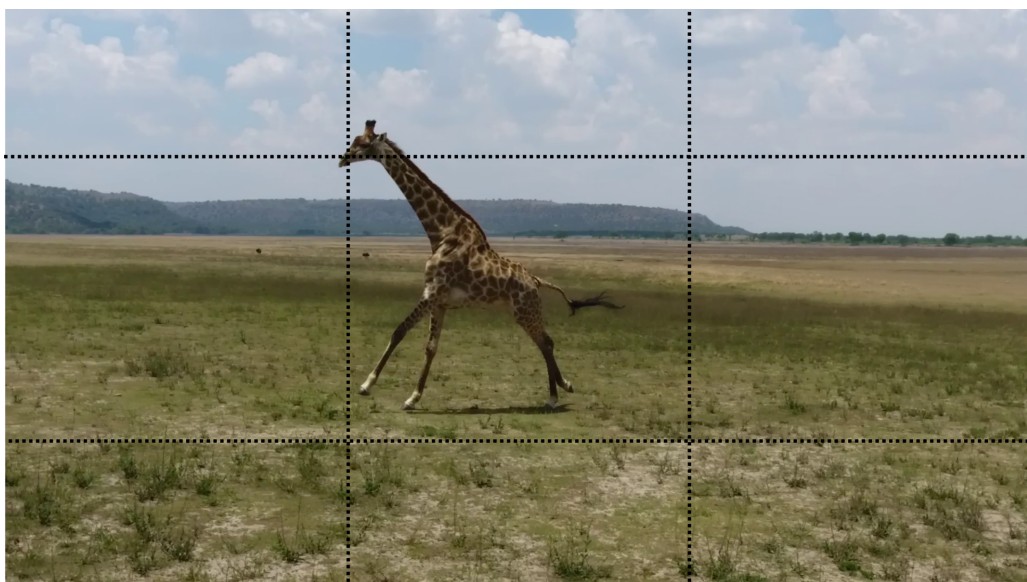

**Figure 8** **A still image of a rotary galloping giraffe taken from video footage recorded using a DJI Phantom 4 UAV, at study site 3.** We recommend that kinematic analyses of the study subject is confined to the centre box of a 3 × 3 grid, overlying the field of view. Linear and angular measurements outside of this area will be subject to greater parallax error. Image source credit C Basu.

subjects. As a general rule, we recommend that subjects are confined at least to the centre block of a 3 × 3 grid in the field of view (Fig. 8), whilst recognising the trade-off between subject focal distance and image resolution.

Whilst the validation study demonstrates the effect of camera position, we cannot directly extrapolate the above measures of speed accuracy to the calculated giraffe kinematics. The measurement accuracy is not solely dependent on the subject focal distance (which could not be measured in this case), but is also dependent on the size of the subject relative to the field of view. For example, a giraffe would encompass a larger proportion of the field of view at 8 m distance than a human subject, and so would be subject to a greater degree of optical error (especially towards the image periphery). In this case, we maintained subject focal distances which similarly positioned the giraffes within the central field of view (Fig. 8), and expect our measurement accuracy to be comparable with the 8% error observed in the laboratory.

When close range photography or artificial ground markers are not feasible, a detailed 3D textured mesh of the terrain may be used to calibrate the images, but this method resulted in an additional calibration error of 5% (Table 2). An error of this magnitude does not alter our findings about giraffe kinematics, but should be considered for future studies. The additional error is related to the subjectivity of picking terrain landmarks which are visible in both the video and 3D mesh. It is important that clearly visible ground features are used; this was illustrated by the difference in error between using artificial ground markers *versus* naturally occurring features (Table 2). A means to address this in future would be to increase the texture resolution of the 3D mesh, to aid in identifying

ground points. This can be done by conducting the aerial survey closer to the ground, but as a trade-off, this demands more images per area, resulting in longer flight times. As UAV battery life improves, this trade-off will become less important.

Camera sampling rate was also a limitation. 120 Hz was the standard for low-cost UAV technology at the time of data collection. Whilst this was sufficient for measuring displacement and speed across the stride, measuring the velocity or acceleration during foot contact events (for the purpose of stride segmentation) resulted in excessive noise. In future, an interpolation approach could be used to artificially increase the sampling rate.

Using this methodology, we were able to measure temporospatial parameters in free-roaming giraffes without any physical contact. We found that giraffes' lack of an intermediate speed gait (e.g., trot/pace) was compensated for by their rotary galloping gait, in that the walk-gallop transition speed of approximately 3.4 ms$^{-1}$ fell close to the mass-specific minimum trotting speed observed in other mammalian quadrupeds (*Heglund & Taylor, 1988*). For example, using Eq. (7), a giraffe weighing 700 kg would be expected to have a minimum trotting speed of 3 ms$^{-1}$.

$$\text{Minimum trotting speed} = 0.593(\text{body mass}^{0.249}). \tag{7}$$

In absolute terms, giraffes can be thought of as being slow gallopers (without routinely using a cantering gait). Beyond the walk-to-gallop transition, increases in galloping speed were almost exclusively achieved by increases in stride length; contrasting with the conserved range of stride frequencies. This pattern is consistent in a wide range of quadrupedal running animals, and is thought to reduce mass-specific energy costs (*Heglund & Taylor, 1988*).

The giraffes in this study galloped with lower duty factors than predicted by dynamic similarity (Table 3). It is tempting to suggest that giraffes experience similarly higher than expected vertical peak ground reaction forces (GRFs) at equivalent speeds, because peak vertical GRF is usually inversely associated with duty factor (*Alexander, 1984*; *Witte, Knill & Wilson, 2004*). Unfortunately in giraffes it may not be possible to accurately predict peak GRF from duty factor alone. In our previous work peak vertical GRF during walking was speed-independent; a finding which may be related to limb compliance (*Basu, Wilson & Hutchinson, 2019*). This could be explored in future with a giraffe musculoskeletal model and a forward dynamics simulation in which peak GRFs are simulated as a function of speed, as experimental data collection from giraffes galloping over force plates may be logistically impossible. Such findings would have implications for muscle power demands and tissue safety factors in giraffes.

Contralateral limb phase decreased with speed (Fig. 7), resulting in a greater degree of overlap between lead and non-lead footfalls. This is consistent with a peak GRF-minimising strategy, where body weight is evenly distributed over two limbs during a greater proportion of the stride. Such a mechanism would be particularly advantageous to giraffes, which possess a disproportionately slender appendicular skeleton, and so may be sensitive to large skeletal compressive and bending stresses (*Biewener, 1983*; *McMahon, 1975*).

The phase relationship between body pitch and neck angle was variable between trials, with a mean congruity of 70%. Given that 100% congruity would represent in-phase neck

ventroflexion and positive body pitching, we speculate that the giraffe neck is inertially stabilised with respect to a world frame, and is effectively decoupled from the motions of the trunk during running; a similar situation to the energy-conserving mechanism observed during walking (*Basu, Wilson & Hutchinson, 2019*). A method of testing this idea in the future would be to examine the effect of ground incline, net acceleration and high-speed turning on neck kinematics. Topography generated by the 3D terrain mapping method used in this study would be particularly useful for this purpose. Alternatively, the variation between strides with respect to phase, as well as the variation in neck angle (Fig. 2B), may indicate an additional or different effect. Variation in angular neck kinematics during galloping has also been noted by *Dagg (1962)*. Defining neck kinematics using a single angle may partly explain this issue, as the cervical vertebral series is far from rigid, and displays varying degrees of curvature over the course of a single stride (Video S1). A spline-based analysis may yield a more robust parameter with which to investigate giraffes' neck kinematics.

The giraffes were studied in their natural habitat, meaning any conclusions can be more confidently applied to giraffes as a wild species, compared with a laboratory setting where conclusions are confined to a specific set of conditions. The drawback is that controlled experimental conditions were not strictly possible. The effects of confounding variables were kept to a minimum by only collecting data from relatively flat terrain, avoiding extremes of weather conditions and comparing giraffes of similar size. Textured terrain models can be used in future to quantify elevation, substrate type and other random effects. Such terrain parameters may useful in investigating giraffes' athletic abilities and energetic costs.

## CONCLUSIONS

This study was a novel application of a UAV system, and has highlighted the gains and technical challenges of this method. We recommend that UAV users minimise kinematic measurement error by maximising the focal distance and confining the study subject to the centre of the field of view. Giraffes' lack of an intermediate gait was compensated for by their rotary galloping gait; giraffes are slow gallopers. Duty factors were lower than predicted by dynamic similarity, suggesting that galloping giraffes may experience high peak ground reaction forces. However, a speed-dependent reduction in contralateral limb phase, and modest maximal speed may maintain appropriate tissue safety factors.

## ACKNOWLEDGEMENTS

We thank Woodlands Hills Wildlife Estate, the Free State Nature Conservation (FSDETEA) and Mangaung Municipality for kindly permitting us to conduct this study on their properties. We appreciated the help of reserve managers William Killian and Rudi Virtue, as well as Ellen Holding for their assistance during the fieldwork. We thank the Natural Wildlife Bridge (Texas, USA), the Highlands Nature Club (Eastern Free State, South Africa), and the University of the Free State (South Africa) for providing logistical support. Thank you to Alice Morrell and Emily Sparkes for assisting with laboratory-based data collection.

Finally, we thank two reviewers, Carlo Massimo Biancardi and Christofer Clemente, for their helpful comments.

### Funding

This work was funded by NERC (PhD studentship for Christopher Basu; grant no. NE/K004751/1 to John Hutchinson), ERC (Advanced grant AD-G 323041 to Alan Wilson), National Research Foundation (South African project no. V106005 to Francois Deacon), Society for Experimental Biology student travel grant (Christopher Basu), and a Company of Biologists Travelling Fellowship (Christopher Basu). The funders had no role in study design, data collection and analysis, decision to publish, or preparation of the manuscript.

### Grant Disclosures

The following grant information was disclosed by the authors:
NERC: NE/K004751/1.
ERC: AD-G 323041.
National Research Foundation: V106005.
Society for Experimental Biology.
Company of Biologists Travelling Fellowship.

### Competing Interests

John R. Hutchinson is an Academic Editor for PeerJ.

### Author Contributions

- Christopher K. Basu conceived and designed the experiments, performed the experiments, analyzed the data, prepared figures and/or tables, authored or reviewed drafts of the paper, approved the final draft.
- Francois Deacon conceived and designed the experiments, performed the experiments, contributed reagents/materials/analysis tools, approved the final draft.
- John R. Hutchinson conceived and designed the experiments, approved the final draft.
- Alan M. Wilson conceived and designed the experiments, contributed reagents/materials/analysis tools, approved the final draft.

### Human Ethics

The following information was supplied relating to ethical approvals (i.e., approving body and any reference numbers):

This study had ethical approval from the Royal Veterinary College (URN 2016 1538) to carry out the study on giraffes and humans within its facilities and in fieldwork.

### Animal Ethics

The following information was supplied relating to ethical approvals (i.e., approving body and any reference numbers):

Both the Royal Veterinary College (URN 2016 1538) and the University of the Free State, South Africa (UFS-AED2016/0063) provided full approval for this observational research on giraffes.

### Field Study Permissions

The following information was supplied relating to field study approvals (i.e., approving body and any reference numbers):

Field experiments were approved by the Free State Province Department of Economic Development, Tourism and Environmental Affairs (permit number 01/34481).

### Data Availability

The kinematic measurements and statistical analyses (in .xlsx format) are available as Supplementary Files. A video of a representative stride is also available (Video S1).

### Supplemental Information

Supplemental information for this article can be found online at http://dx.doi.org/10.7717/peerj.6312#supplemental-information.

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
