# Peer review of "The running kinematics of free-roaming giraffes, measured using a low cost unmanned aerial vehicle (UAV)"

_PeerJ, doi:10.7717/peerj.6312_

## Round 0.1 · original submission · Minor Revisions

Both reviewers are very positive about your manuscript and have only raised a few points that need addressing - mostly in the introduction (reviewer 1) and methods (reviewer 2). I look forward to seeing a revised version in the near future.

·

Basic reporting

1) Clear, unambiguous, professional English language used throughout.

Excellent. No comments.

2) Intro & background to show context. Literature well referenced & relevant.

The introduction and background are very good. However, I would suggest the authors to consider expanding the part on gait dynamics, from line 74 to 89, including the following information:
According to Hildebrand (1976) and Cartmill (2002), the lateral walking gait of camels, giraffes and other few species differs from the lateral walking gait of others cursorial species by one parameter, which is diagonality (i.e. “% of stride interval that footfall of the forefoot follows hind on the same side” – Hildebrand, 1976). Depending on diagonality, the lateral walking sequence can follow a “Horse rule”, which will develop in a trot at running speed, or a “Camel rule”, which will develop in a pace at running speed (Cartmill et al., 2002). The lateral walk of giraffes follows the “Camel rule” (Cartmill, 2002, reporting observations of Hildebrand).
The diagonality (sensu Hildebrand and Cartmill) is equivalent to the phase difference explained by Alexander and Jayes in the figure 2 of their cited work of 1983 (Although slightly different in the calculation). In Alexander and Jayes (1983) only camels lies on the line “lateral walk to pace”, giraffes were not included in their sample.

I would suggest changing the sentence that begins at line 81. According to Hildebrand and Cartmill (cited above), and confirmed by a robot model simulation (Suzuki et al., 2016), the intermediate speed gait of giraffes should be pace.

- Cartmill, M., Lemelin, P., & Schmitt, D. (2002). Support polygons and symmetrical gaits in mammals. Zoological Journal of the Linnean Society, 136(3), 401-420.
- Suzuki, S., Owaki, D., Fukuhara, A., & Ishiguro, A. (2016, July). Quadruped gait transition from walk to pace to rotary gallop by exploiting head movement. In Conference on Biomimetic and Biohybrid Systems (pp. 532-539). Springer, Cham.

Literature is relevant and well referenced. I have found only a little discrepancy.
Line 73: The cited work is missing in the reference list: Biancardi and Minetti, 2012
Biancardi, C. M., & Minetti, A. E. (2012). Biomechanical determinants of transverse and rotary gallop in cursorial mammals. Journal of Experimental Biology, jeb-073031.

3) The structure of the submitted article should conform to one of the templates.

No comments.

4) Figures should be relevant to the content of the article.

The figures are good and relevant.

5) The submission should be 'self-contained,' should represent an appropriate 'unit of publication,' and should include all results relevant to the hypothesis.

No comments.

6) All appropriate raw data has been made available in accordance with our Data Sharing policy.

No comments. Thank you for providing raw and supplementary data.

Experimental design

1) Original primary research.

No comments.

2) Research question well defined, relevant & meaningful. It is stated how research fills an identified knowledge gap.

Excellent. Besides the methodological relevant question of kinematic data collection in field, the problem of the giraffe locomotion is interesting, with different knowledge gaps to fill.

3) The investigation must have been conducted rigorously and to a high technical standard.

Excellent

4) Methods described with sufficient detail & information to replicate.

Excellent. The methodology is extensively explained. Calibration, data processing and validation of the methodology are very useful.

5) The research must have been conducted in conformity with the prevailing ethical standards in the field.

No comments.

Validity of the findings

1) The data should be robust, statistically sound, and controlled.

No comments.

2) The data on which the conclusions are based must be provided or made available in an acceptable discipline-specific repository.

No comments.

3) The conclusions should be appropriately stated, should be connected to the original question investigated, and should be limited to those supported by the results.

Excellent

4) Speculation is welcomed, but should be identified as such.

No comments.

5) Decisions are not made based on any subjective determination of impact, degree of advance, novelty, being of interest to only a niche audience, etc. Replication experiments are encouraged (provided the rationale for the replication, and how it adds value to the literature, is clearly described); however, we do not allow the ‘pointless’ repetition of well known, widely accepted results.

No comments.

6) Negative / inconclusive results are acceptable.

No comments

Additional comments

This is an excellent paper, well written, with, at least, two different and useful results:
1) The validation of a kinematic measurement system based on a drone (UAV) that carries a high speed camera. This is a good example about how low-cost new technologies can be employed to produce amazing science.
2) The kinematic gait analysis of giraffes in field. Even if there is a lot of footage of free ranging animals available, in more or less public repositories (BBC, Arkive, You tube), the quality of those videos is often questionable and problematic.
The accurate analyses of the data provided new interesting information about the locomotion of these amazing species. Giraffes are anomalous rotary gallopers, “slow gallopers”, as the authors reported. From a visual inspection, their gallop appears different to that of a cheetah or a gazelle, despite of the same footfall pattern. These sound quantitative results fill a long-standing knowledge gap.

·

Basic reporting

no comment

Experimental design

no comment

Validity of the findings

no comment

Additional comments

This was an interesting and useful paper to read on the validity of using UAV's to measure biomechanics in the field. I think many researchers will find this to be of general interest. The information gathered on giraffe locomotion is also of interested to a broard range of biomechanists.

The paper itself was very well written, clearly describing the steps undertaken. I have only a few minor comments, the most important being the influence of focal distance on speed accuracy estimates, and why so few test distances were used.

Abstract
Line 18: Reference to the GPS + inertial sensors seems out of place here, makes more sense in the intro.
Line 30: The gold standard was a bit ambiguous in this sentence, maybe define more clearly how it was measured and report % errors.

Methods

Line 109: The mean, max and min focal distances used in the field should be reported. This could also be included as a random factor in a linear mixed effect model. (possibly along side subject).

Line 137: Video calibration. Was any form of data smoothing used to reduce digitizing error? Was this source of error measured?
Line 203: ‘’user-based determination of footfalls’’ not really sure what is meant by this. Does it mean no one limb was used to define the start of a footfall sequence?
Line 285: How much lower were speeds estimated by UAV’s? Was this reduction reflected in the percent error reported below? Would be good to get an estimate of how much lower speeds were (i.e. speeds were 90% of treadmill speed). Also good to know how repeatable this was (i.e. 90% + / - 5% lower). This would be useful for anyone using the speed estimates derived from UAV’s – i.e. they might consider increasing the reported speed by the error reported.

Line 288: 8% error at 8m focal distance. How would this translate to the 40+ m focal distance used during field trials. Surprised to see so few test distances are used here considering that these are relatively easy to collect in a lab setting. Further these tested distances do not seem to include the actual distances used in the field. Do we expect error to consistently be reduced with distance? Or will it flatten out and become constant at some distance?

---

## Round 0.2 · accepted · Accept

Thanks for attending to all the reviewers' comments so thoroughly, and thanks for making such good use of the PeerJ Preprints system as well. I am satisfied that all the comments have been dealt with appropriately and I'm now very happy to recommend this article for publication. I'll look forward to seeing the final published version. Congratulations!

#